# Influence of Epilepsy Characteristics on the Anxiety Occurrence

**DOI:** 10.3390/brainsci14090858

**Published:** 2024-08-26

**Authors:** Aleksandar Gavrilovic, Jagoda Gavrilovic, Jelena Ilic Zivojinovic, Ljiljana Jeličić, Snezana Radovanovic, Katarina Vesic

**Affiliations:** 1Department of Neurology, Faculty of Medical Sciences, University of Kragujevac, 34000 Kragujevac, Serbia; aleksandar.gavrilovic@fmn.kg.ac.rs (A.G.); katarina.vesic@fmn.kg.ac.rs (K.V.); 2Clinic of Neurology, University Clinical Center Kragujevac, 34000 Kragujevac, Serbia; 3Department of Infectious Diseases, Faculty of Medical Sciences, University of Kragujevac, 34000 Kragujevac, Serbia; 4Clinic for Infectious Diseases, University Clinical Center Kragujevac, 34000 Kragujevac, Serbia; 5Faculty of Medicine, Institute of Hygiene and Medical Ecology, University of Belgrade, 11000 Belgrade, Serbia; jelena.ilic-zivojinovic@med.bg.ac.rs; 6Cognitive Neuroscience Department Research and Development Institute “Life Activities Advancement Institute”, 11000 Belgrade, Serbia; lj.jelicic@add-for-life.com; 7Department of Speech, Language and Hearing Sciences, Institute for Experimental Phonetics and Speech Pathology, 11000 Belgrade, Serbia; 8Department of Social Medicine, Faculty of Medical Sciences, University of Kragujevac, 34000 Kragujevac, Serbia; snezana.radovanovic@fmn.kg.ac.rs

**Keywords:** epilepsy, anxiety, seizure, drug responsivity, drug sensitivity, drug resistance

## Abstract

The presence of anxiety in individuals with epilepsy significantly influences their medical treatment and quality of life and often goes unrecognized or untreated, posing a challenge to differential diagnosis. The study aimed to investigate the influence of epilepsy characteristics on anxiety. The research involved 155 patients with generalized and focal drug-sensitive [DSE] and drug-resistant [DRE] epilepsy. Hamilton anxiety rating scale [HAS] was used to assess the symptoms of anxiety at three time points [baseline, 12, and 18 months]. DSE patients exhibited significantly lower HAM-A scores than patients with DRE at the initial visit [*p* = 0.000] after 12 [*p* = 0.000] and 18-month follow-up [*p* = 0.000]. Focal DRE patients presented higher HAM-A scores than focal DSE patients in the initial visit [*p* = 0.000] after 12 [*p* = 0.000] and 18 months [*p* = 0.000]. Medication responsiveness, seizure type, and illness duration emerged as significant anxiety predictors [*p* = 0.000]. After 18 months of follow-up, significant contributors to anxiety were drug responsivity and illness duration [*p* = 0.000]. The occurrence of anxiety in epilepsy patients is most significantly influenced by well-controlled epilepsy and a positive response to medication.

## 1. Introduction

Epilepsy is a chronic neurological condition frequently accompanied by other significant neurological, cognitive, and psychosocial consequences [1]. Psychiatric comorbidities are present in patients with epilepsy (PWE) two to three times stronger than in the general population [2]. The understanding of the relationship between epilepsy and psychiatric comorbidities remains limited. Nevertheless, evidence suggests that PWE face significant risks due to psychosocial factors, structural abnormalities, and electrophysiological irregularities, particularly within the limbic system [3,4,5,6]. The use of anti-seizure medications, some with adverse psychotropic effects, and psychotropic drugs, which can reduce the seizure threshold, further complicates the clinical landscape [7]. This intertwining of epilepsy and psychiatric disorders creates a complex scenario with considerable consequences for overall treatment outcomes. Poor adherence to anti-seizure medications, increased seizure frequency, and a less favorable epilepsy treatment outcome are all associated with psychiatric comorbidities [8].

According to one comprehensive Canadian study, all mental disorders were found to be 35.5%, with depression, anxiety, and psychotic disorders emerging as the most prevalent [9,10]. A total prevalence of 20.2% of anxiety disorders is present in PWE, according to recent meta-analysis results [2].

There is a bidirectional relationship between anxiety and epilepsy, possibly involving common underlying mechanisms [11,12]. Dysfunction of neurotransmitter systems, dysregulation of the hypothalamic-pituitary-adrenal (HPA) axis, structural changes, or chronic neuroinflammation may be involved in the pathophysiology of both conditions as biological mechanisms [13,14,15,16]. The lack of genetic overlap between epilepsy and psychiatric disorders has already been pointed out [17].

There has been ongoing discourse regarding the various types of anxiety in PWE, exploring their pathophysiology and clinical manifestations. These distinctions extend beyond their temporal relation to seizures and encompass differences between subjective quality and behavioral consequences [11,18]. Therefore, anxiety represents a wider spectrum of manifestations, not individual comorbidity [19]. This range incorporates diverse pathogenic mechanisms, including pre-ictal prodromes potentially symptomatic of ictal excitability changes, direct neurobiological mechanisms linked with the engagement of brain structures that are involved in emotional perception and regulation, early ictal anxiety, which reflects loss of control and inter-ictal anxiety, such as anticipation of potential future seizures, together with phobic behavior as a manifestation of social stigma [20,21].

In the present study, we try to examine, at the same time, the anxiety prevalence in individuals with chronic DSE and DRE patients at three different time points. The primary objective is to compare the severity of anxiety in both patient groups and examine whether the frequency of anxiety changes during a follow-up period of 18 months. Furthermore, the study aims to investigate potential correlations between demographic and clinical characteristics and anxiety in PWE. In addition, we investigate and identify the possible clinical variables that affect the occurrence of anxiety over time and explore whether they change during the 18-month follow-up period.

## 2. Methods

### 2.1. Study Population

This study presents an additional analysis of data derived from a prior study that investigated the cognitive status of patients with drug-sensitive and drug-resistant epilepsy, including both generalized and focal types [22]. Conducted at the Clinic for Neurology, Clinical Center in Kragujevac, Serbia, the study enrolled 155 patients diagnosed with epilepsy. Two neurologists with experience in epilepsy revised the diagnostic criteria during their scheduled visits. Inclusion criteria required participants to be over 18 years old and diagnosed with epilepsy by experienced epileptologists according to the International League Against Epilepsy [ILAE] criteria [23,24]. Exclusion criteria involved severe chronic medical conditions, acute infections, recent surgeries, alcohol or drug abuse, psychogenic non-epileptic seizures, and comorbidities of mental conditions, such as psychosis, dementia, and mental retardation. The classification of epilepsy into drug-sensitive or drug-resistant categories was based on ILAE recommendations and therapy response. Drug-sensitive epilepsy (DSE) was defined as effectively controlled by anti-seizure drugs. Drug-resistant epilepsy (DRE) was defined as the persistence of seizures despite at least two syndrome-adapted anti-seizure drugs (ASD) used at efficacious daily doses, whether as monotherapies or in combination, to achieve sustained seizure freedom [25]. According to this classification, patients were divided into two main groups: DSE and DRE groups. In addition, using ILAE criteria, patients are divided into four subgroups: drug-sensitive generalized (DSG), drug-sensitive focal (DSF), drug-resistance generalizes (DRE), and drug-resistance focal (DRF) patient subgroups, including both generalized and focal epilepsy types. The study received approval from the Ethics Committee of the relevant institution (Ethics Committee report number 01-5412, dated 1 June 2017, and Ethics Committee decision number 01-2234, dated 11 January 2020), and written informed consent was obtained from all participants.

### 2.2. Neurological Assessment

The patient’s basic characteristics—age, sex, body mass index, and illness duration—are determined based on anamnesis or hetero-anamnesis and a sociodemographic questionnaire. The frequency and severity of seizures were assessed based on the seizure diary. Patients were provided with a dedicated seizure journal for the research, recording daily therapy and potential seizure occurrences. The severity of the neurophysiological correlate was measured based on the electroencephalogram. Each patient underwent an electroencephalogram [EEG], with EEG findings categorized as regular, nonspecific, non-epileptiform-altered, or epileptiform-altered. Applied therapy and its side effects are also based on auto-anamnesis data. Compliance with recommended pharmacotherapy was evaluated based on the information documented in the journal. The neurological assessment, somatic examinations, vital signs, and neurological and psychological status were conducted at the baseline visit, 12 months, and 18 months after the study’s initiation.

### 2.3. Psychological Assessment

The psychological assessment incorporated Hamilton’s Anxiety Assessment Scale [HAM-A] to evaluate the existence and intensity of anxiety [26]. Comprising fourteen items, each independently scored on a five-point ratio scale, and HAM-A scores span from 0 to 56. A score of 17 or below indicates mild anxiety, scores ranging from 18 to 24 denote mild to moderate anxiety, and scores within the range of 25 to 30 signify moderate to severe anxiety.

### 2.4. Statistical Analysis

Statistical analyses were performed using SPSS 20.0 software (SPSS Inc, Chicago, IL, USA). Descriptive statistics presented categorical variables as absolute (n) and relative frequencies (%), and continuous variables were reported as mean, standard deviation (SD), standard error (SE), and percentages. The normal distribution of data was assessed using the Shapiro–Wilk test. For continuous variables, the nonparametric Mann–Whitney U or Kruskal–Wallis test (or analysis of variance, when appropriate) was employed to examine differences in HAM-A scores between two or more epilepsy patient groups, while the Chi-square test was used to assess differences in sociodemographic characteristics. Friedman’s test was utilized to assess variations in HAM-A scores between baseline, 12, and 18-month follow-up. Spearman rank correlation analysis was used to examine the correlation between anxiety symptoms and drug responsivity, epilepsy type, and illness duration. The same test is used to assess the correlation between the degree of anxiety and seizure control. Correlation coefficients between 0.5 and 1 were considered strong, between 0.3 and 0.5 moderate, and less than 0.3 weak. Multiple regression analysis was conducted to explore significant sociodemographic and clinical predictors of HAM-A scores. To investigate the optimal prediction of the presence of anxiety symptoms at the three time points (baseline, after 12 and 18 months), we performed three separate binary logistic regression analyses. HAM-A scores were used to assess the presence and severity of anxiety with cutoff values set at <14 to discriminate patients with and without anxiety. A *p*-value of 0.05 was deemed statistically significant. The sample size was estimated according to data on the value of the correlation coefficient from a study that investigated the correlation of quality of life in patients with pharmacoresistant and pharmacosensitive epilepsy and psychogenic non-epileptic seizures, which is r = 0.53 [27]. Taking alpha as 0.05 and the power of the study of 0.8, the correlation sample was calculated according to the statistical program G Power 3. In this way, the expected sample size of 75 patients in each group was calculated, with 75 pharmacoresistant epilepsy and 75 pharmacosensitive, the total number of 150 subjects. Considering a significant percentage difference in the prevalence of different clinical forms of the disease, our planned groups will have 50 subjects with pharmacosensitive and 100 subjects with pharmacoresistant epilepsy.

## 3. Results

### 3.1. Patients Sociodemographic and Clinical Data

Included in this study were 155 patients diagnosed with epilepsy. Among these, 103 patients [66.5%] exhibited drug-resistant epilepsy [DRE], while 52 patients [33.5%] had drug-sensitive epilepsy [DSE]. Focal seizures were identified in 84 [54.2%] patients, while 71 [45.8%] presented with generalized seizures. In the DSE group, 23 [44.2%] patients had generalized epilepsy, and 29 [55.8%] had focal epilepsy. Specifically, 17 patients [58.6%] had temporal, 11 patients [37.9%] had frontal, and one patient [3.4%] had occipital epilepsy. For those with DRE, 48 patients [46.6%] had generalized epilepsy, and 55 [53.4%] had focal epilepsy. Among patients with focal epilepsy in the DRE group, 33 [59.3%] had temporal, 16 [29.8%] frontal, 2 [3.6%] parietal, and 4 [7.3%] patients had occipital epilepsy. Statistically significant differences between the groups in terms of gender, age, and illness duration were not observed. Further clinical assessment details can be found in Table 1 [24].

### 3.2. Higher HAM-A Score in Patients with DRE

The Friedman test demonstrated a noteworthy increase in HAM-A scores after 12 and 18 months from the commencement of the follow-up [17.96 ± 10.32 vs. 19.23 ± 12.05 vs. 21.23 ± 13.75; Chi-Square [2/155] = 75.220, *p* = 0.000]. Notably, patients with drug-sensitive epilepsy [DSE] exhibited significantly lower HAM-A scores compared to patients with drug-resistant epilepsy [DRE] at the initial visit [2.04 ± 3.34 vs. 12.95 ± 11.33, *p* = 0.000], after 12 months [2.46 ± 4.24 vs. 17.45 ± 13.23, *p* = 0.000], and at the 18-month follow-up [5.15 ± 5.07 vs. 21.23 ± 13.75, *p* = 0.000] [see Figure 1].

The HAM-A scores exhibited significant variations among different types of epilepsy. In the initial visit, patients with focal drug-resistant epilepsy [DRE] presented higher HAM-A scores compared to those with focal drug-sensitive epilepsy [DSE] [14.64 ± 11.69 vs. 1.66 ± 3.03, *p* = 0.000]. After 12 months, the HAM-A scores remained elevated in patients with focal DRE in comparison to those with focal DSE [17.58 ± 13.87 vs. 2.24 ± 4.61, *p* = 0.000]. Even at the 18-month mark, patients with focal DRE maintained higher HAM-A scores than those with focal DSE [20.89 ± 15.35 vs. 2.76 ± 5.22, *p* = 0.000]. This group exhibited the highest degree of anxiety and continued to differ significantly from both generalized DSE and DRE. No significant difference was observed between the last two types of epilepsy [Figure 2].

### 3.3. The Influence of Illness Duration, Drug Responsivity, and Epilepsy Type on Anxiety in Patients with Drug-Resistant Epilepsy

We identified a significant positive correlation between anxiety symptoms and drug responsivity [r = 0.40, *p* = 0.000], epilepsy type [r = 0.30, *p* = 0.001], and illness duration [r = 0.30, *p* = 0.000]. Conversely, a noteworthy negative correlation was observed between the degree of anxiety and seizure control [r = −0.40, *p* = 0.000]. Furthermore, responsiveness to medication [referred to as DSE or DRE], the type of epileptic seizures, and illness duration emerged as significant predictors of the Hamilton anxiety scale [HAS] score [F [9/82] = 4.590, *p* = 0.000, R^2^ = 0.361, adjusted R = 0.283] [see Table 2].

Furthermore, we observed a progressive increase in the proportion of patients with anxiety over time [18.2% vs. 22.6% vs. 27.7%]. Binary logistic regression demonstrated a significant association between the presence of anxiety at baseline and EEG findings [χ^2^ [9/151] = 42,870, *p* = 0.000]. Subsequently, a significant relationship between drug responsivity, epilepsy type, epilepsy duration, and anxiety was identified after 12 months [χ^2^ [9/151] = 49,286, *p* = 0.000]. By the 18-month mark, the most significant contributors to the presence of anxiety were drug responsivity and illness duration [χ^2^ [9/151] = 50,915, *p* = 0.000] [Table 3].

## 4. Discussion

The study point of the present research was to compare the degree of anxiety severity between two groups of patients, with DSE and DRE, as well as to determine predictive values of factors related to epilepsy and demographic factors.

DRE patients exhibited significantly higher HAM-A scores compared with DSE patients. Throughout the follow-up period, the highest levels of anxiety were consistently observed in patients with focal DRE. Positive correlations were identified between anxiety symptoms and factors such as drug responsivity, epilepsy type, and illness duration, while a negative correlation was noted between anxiety and seizure control. Additionally, responsiveness to medication, the type of epileptic seizures, and illness duration emerged as significant predictors of HAS score. During the research, we noticed a significant increase in anxiety in patients. Notably, after 18 months, the primary contributors to the presence of anxiety were identified as drug responsivity and illness duration.

Prior research indicates that 20–30% of PWE experience psychiatric comorbidities [28]. The prevalence of these comorbidities varies depending on the response to treatment and type of epilepsy [29]. Refractory epilepsy is recognized as a risk factor for both depression and anxiety. Existing evidence suggests that individuals with epilepsy who endure a chronic, refractory seizure disorder are more prone to experiencing mood disorders and symptoms of anxiety [30,31,32,33]. Our findings align with previous research because we revealed significantly higher HAM-A scores in DRE patients compared to DSE patients. Conversely, in the investigation conducted by Gugała-Iwaniuk et al. (2021), the levels of anxiety symptoms did not show a significant difference between drug-resistant and drug-sensitive patient groups. Furthermore, pharmacoresistance demonstrated a significant association with the severity of depression and anxiety in male patients [34].

All research regarding the lateralization of epilepsy, localization, and type of epilepsy on the one hand and anxiety on the other showed inconsistency [35,36,37]. Notably, a recent study identified a distinctive association between severe anxiety symptoms and focal or unknown epilepsy types, as opposed to generalized epilepsy types [35]. In the present study, the highest levels of anxiety were consistently observed in patients with focal DRE. Most of the patients had temporal and frontal lobe epilepsy. The opinion of certain authors is clear that patients with focal, especially temporal epilepsy of the dominant hemisphere, are primarily affected by depressive disorders in epilepsy [38]. Conversely, other investigations have indicated similar occurrences of depression and anxiety in patients with right or left temporal lobe epilepsy, suggesting a potential connection with pathological hyperexcitability in both hemispheres [31,39]. A recent study indicated a significant improvement in these symptoms following anterior temporal lobectomy [40,41]. About 15% of all patients with pharmacoresistant epilepsy have frontal lobe epilepsy [42]. It is associated with various psychopathological manifestations, such as executive dysfunctions, addictive behavior, cognitive impairment, hyperactivity, and obsession [43]. Tang et al. demonstrated that patients with frontal lobe epilepsy exhibited a significantly higher degree of anxiety compared to those with generalized epilepsy [44].

Epileptic activity in specific brain regions can directly trigger paroxysmal anxiety, often presenting as panic attacks [45]. This has given rise to a theory proposing a shared pathophysiological mechanism for both anxiety attacks and epilepsy, predominantly centered in the amygdala [46]. It is noteworthy that ictal anxiety is also linked with seizures originating in the frontal lobe, specifically the anterior cingulate or orbitofrontal cortex. The hypothesis implies that the anterior cingulate plays a regulatory role in amygdala activity, and this regulatory function may be abnormal in individuals with anxiety [47,48]. Decreased anterior cingulate activation in patients with generalized anxiety disorder was shown by the results of functional MRI studies [49]. In recent years, accumulating evidence indicated that inflammatory pathways and neuroinflammation have a significant role in the pathophysiology of epilepsy and psychiatric disorders and that neuroinflammation can be a link between epilepsy and depression, anxiety, and cognitive impairment [50,51]. Previous studies investigate elevated levels of different pro-inflammatory cytokines and other inflammatory markers in serum and cerebrospinal fluid in PWE. Neuroinflammation can alter microglia and asrocyte function, increase excitability, cause epileptogenesis, and change neuroplasticity [16]. Some investigators pointed out that interictal cytokines levels depend on the severity and frequency of seizure [52,53]. In addition, long-lasting smoldering inflammation leads to neurodegeneration [16]. These mechanisms can trigger depressive and anxiety disorders in PWE. A recent study evaluated the course of newly diagnosed epilepsy in adult patients with infective etiology. The authors estimated that infection etiology (neurocysticercosis) was associated with DSE, while structural changes in the brain were related to seizure occurrence, and the smaller size of lesions contributed to better treatment response [54].

Increasing epidemiological evidence suggests the existence of common risk factors influencing both anxiety and seizure susceptibility. Consistently identified factors associated with anxiety across studies include female gender, depression, and unemployment [35]. However, results for other demographic and epilepsy-related factors vary significantly among studies, and the majority had limited power to detect associations due to small sample sizes. We found a positive correlation between anxiety symptoms and drug responsivity, epilepsy type, and illness duration. Additionally, responsiveness to medication, the type of epileptic seizures, and illness duration were marked as significant predictors of HAS score. At the conclusion of an 18-month follow-up, the factors most strongly predicting the presence of anxiety in relation to epilepsy were recognized as drug responsivity and illness duration. Epilepsy type, female gender, and high seizure frequency are separate, positively associated factors for the development of anxiety and/or depression [35]. In many quality investigations, a connection between psychological disorders and seizures has been observed [55,56,57,58]. Late seizure onset and stigma are potential risk factors for anxiety in patients with epilepsy [59,60].

This study has some limitations. First, we included only patients with chronic and long-lasting epilepsy and with heterogeneous epilepsy etiology. Second, the study was limited in terms of anti-seizure medications due to a lack of data on the drug concentration in the blood. Third, the follow-up period in the present study was short, only 18 months.

## 5. Conclusions

Our study emphasizes the high prevalence of anxiety in patients with epilepsy, particularly among individuals with drug-resistant epilepsy, at three different time points. In addition, results implicated that anxiety tends to increase during a follow-up period of 18 months. Drug responsivity, type, and duration of epilepsy are the most significant factors that have an impact on anxiety symptoms in a short period of follow-up, while seizure control is a significant factor in a longer follow-up period (18 months). The previous studies confirm the frequent occurrence of psychiatric disorders in people with epilepsy, significantly impacting their quality of life. Recognizing the significance of these comorbidities, clinicians must possess the skills to identify and manage them, incorporating routine anxiety screening tools. Given the critical nature of this issue, ongoing initiatives are imperative to monitor and address the prevalence and impact of anxiety in epilepsy patients over a long period. Implementing these measures is crucial, not only for improving the quality of life for patients with epilepsy but also for optimizing the effectiveness of the treatment.

## Figures and Tables

**Figure 1 brainsci-14-00858-f001:**
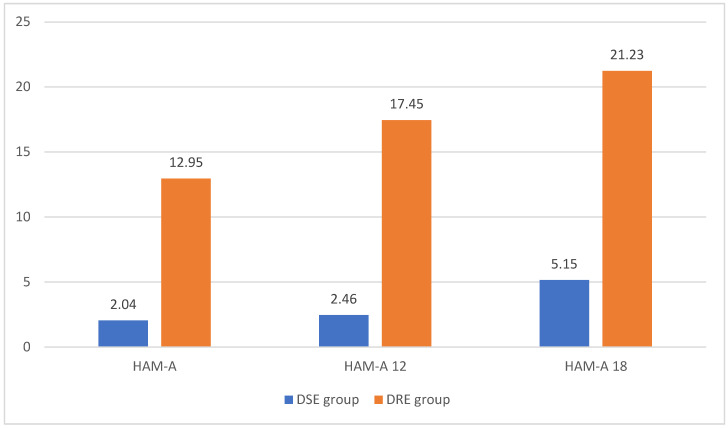
The HAM-A score at baseline, after 12 months, and after 18 months in both drug-sensitive and drug-resistant epilepsy patient groups. Abbreviations: HAM-A—Hamilton’s Anxiety Assessment Scale, DSE—Drug-Sensitive Epilepsy, DRE—Drug-Resistant Epilepsy, *p* < 0.05.

**Figure 2 brainsci-14-00858-f002:**
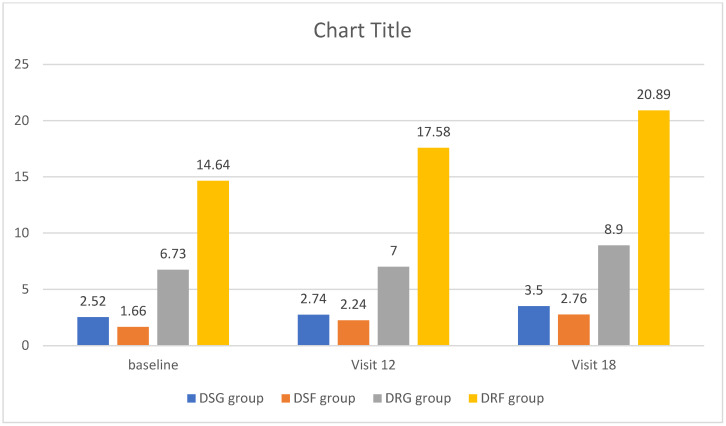
Anxiety occurrence in different epilepsy patient groups. Abbreviations: HAM-A—Hamilton’s Anxiety Assessment Scale, DSG—Drug-Sensitive Generalized, DSF—Drug-Sensitive Focal, DRG—Drug-Resistant Generalized, DRF—Drug-Resistant focal, *p* < 0.05.

**Table 1 brainsci-14-00858-t001:** Demographic and clinical features in patients with epilepsy.

Variables	DSE Group	DRE Group	*p*
Age (years)	38.18 ± 16.43	35.61 ± 12.52	*p* = 0.659
Sex (male/female)	27/25	50/53	*p* = 0.820
Mean duration in illneses (years)	9.56 ± 7.31	10.48 ± 758	*p* = 0.459
Genelalized epilepsy	23 (44.2%)	48 (46.6%)	*p* = 0.913
Focal epilepsy	29 (55.8%)	53 (53.4%)	
Temporal	17 (58.6%)	33 (59.3%)	
Frontal	11(37.9%)	16 (29.8%)	
Parietal	/	2 (3.6%)	
Occipital	1 (3.4%)	4 (7.3%)	
Poor seizure control	/	25 (24.3%)	
Medium seizure control	/	63 (61.2%)	
Good seizure control	12 (23.1%)	9 (8.7%)	
Rare seizures	12 (23.1%)	5 (4.9%)	
Remmision	28 (53.8%)	1 (0.9%)	
Monotherapy	52 (100)	/	
Duotherapy	/	73 (70.9%)	
Polytherapy	/	30 (29.1%)	
Topiramate	5 (9.6%)	23 (23.3%)	
Levetiracetam	20 (38.5%)	97 (94.2%)	
Carbamazepine	12 (23.1%)	44 (42.7%)	
Phenobarbiton	2 (3.8%)	10 (9,7%)	
Lamotrigine	7 (13.5%)	28 (27.2%)	
Valproat acid	6 (11.5%)	33 (32.%)	
Seizure freedom > 1 month	52 (96.2%)	9 (8.7%)	
Seizure freedom 10 days > month	2 (3.8%)	43 (41.7%)	
Seizure freedom < 10 days	/	51 (49.5%)	

Abbreviations: DSE—Drug-Sensitive Epilepsy, DRE—Drug-Resistant Epilepsy, *p* < 0.05.

**Table 2 brainsci-14-00858-t002:** The significant predictors for HAM-A score.

Explanatory	Unstandardized	Standardized	t	*p*
Variables	Coefficient ± SE	Coefficient		
Drug responsivity	13.369 ± 6.644	0.460	2.012	0.048 *
Type of epilepsy	−3.122 ± 1.620	−0.185	−1.927	0.058 *
Illness duration	−0.607 ± 0.199	−0.303	3.048	0.003 *
Seizures control	1.034 ± 2.219	0.104	0.466	0.642
EEG findings	−1.239 ± 0.17	−0.103	−1.014	0.314

Abbreviations: EEG—Electroencephalography, *p* < 0.05, *—statistically significant.

**Table 3 brainsci-14-00858-t003:** The occurrence of anxiety at three time points is shown as the dependent variable, and clinical data as explanatory variables.

Categories	Explanatory Variables	B	SE	Wald	*p*	95% CI
Anxiety	EEG findings	0.716	0.366	3.826	0.05	0.99–4.912
Baseline						
Anxiety 12	Drug responsivity	−5.169	1.991	6.738	0.009	0.000–0.282
	Epilepsy duration	0.126	0.051	6.064	0.014	0.798–0.975
	Type of epilepsy	0.021	0.525	3.780	0.052	0.992–7.763
Anxiety 18	Drug responsivity	−5.131	1.906	7.250	0.007	0.00–0.248
	Epilepsy duration	−0.139	9.053	6.980	0.008	0.785–0.965
	Type of epilepsy	0.588	0.411	2.502	0.152	0.805–4.028
	EEG findings	−0.045	0.333	0.018	0.892	0.498–1.834
	Seizure control	−0.336	0.606	0.308	0.579	0.218–2.341

Abbreviations: EEG—Electroencephalography, *p* < 0.05.

## Data Availability

Data is contained within the article.

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
