# Peer review of "Influence of Epilepsy Characteristics on the Anxiety Occurrence"

_brainsci, 2024, doi:10.3390/brainsci14090858_

Round 1

Reviewer 1 Report

Comments and Suggestions for Authors

The presence of anxiety disorders in patients with epilepsy is a well-known problem and has a completely logical explanation. First of all, this is due to the constant fear of a new seizure, which is especially typical for patients with focal forms of epilepsy, since these patients remember their seizures. It is quite obvious that patients in remission from epilepsy have a lower level of anxiety, since these patients do not have a fear of a sudden onset of an epileptic seizure. In this regard, the conclusions obtained by the authors are quite obvious and have already been confirmed many times earlier. The authors of the manuscript indicate that a study of this type is being conducted for the first time, but this contradicts the "discussion" section of the manuscript, in which they compare their results with the results of previous studies. Thus, it makes sense to indicate what exactly is unique and novel about this study. Also, the manuscript does not contain information on what antiepileptic therapy was received by patients in both groups. This is important, since some antiepileptic drugs have an anti-anxiety effect, while some, on the contrary, can increase anxiety. It is also important to indicate the number of drugs used by patients (monotherapy, duotherapy, polytherapy), the concentration of these drugs in the blood. Also, the manuscript does not say anything about the frequency of seizures in the observed patients, this can also affect the presence of anxiety and its severity. The question also arises about the presence of depressive disorders in these groups of patients, did the authors of the manuscript take this into account?

Reviewer 2 Report

Comments and Suggestions for Authors

The current manuscript formant is not according to the “Instruction for authors.” Please, dear authors review the instructions and modify accordingly, otherwise there are several missed pieces of information for evaluation of the manuscript quality.

Could the authors provide information of how their study differ from other from the literature?

Write the IRB approval number at the methodology and at the end of the study after conclusion.

How did the authors define DRE? Include this information in the main text.

Did two neurologists diagnose all the patients? Include this information in the main text.

Was requested permission to use HAM-A? Did the authors hold the copyrights for its use?

Please provide full description of SPSS.

How was calculated the power of the study?

How were the variables assessed? Please describe about variables were used with the specific statistical methods.

What were the antiseizure medications in use by the patient? How many? Please include this data in the table, it is important for the definition and characterization of the population in the current study.

Please remove the term “antiepileptic drug,” which should be modified to antiseizure medications.

The authors did not describe in the methodology the division between groups, but at the results they present different groups. Please provide information in the methodology regarding this.

Review table 2 and 3 structure, the numbers are superimposed.

In the discussion other things that can be added is the relationship with anxiety and seizure, and inflammatory markers Arend J, Kegler A, Caprara ALF, Almeida C, Gabbi P, Pascotini ET, de Freitas LAV, Miraglia C, Bertazzo TL, Palma R, Arceno P, Duarte MMMF, Furian AF, Oliveira MS, Royes LFF, Mathern GW, Fighera MR. Depressive, inflammatory, and metabolic factors associated with cognitive impairment in patients with epilepsy. Epilepsy Behav. 2018 Sep;86:49-57. doi: 10.1016/j.yebeh.2018.07.007. Epub 2018 Aug 2. PMID: 30077908.

Other feature that can be discussed is the fact of DRE and the etiology is likely structural, and since it can be infectious there is the correlation of infectious disease and neuropsychological features. Caprara ALF, Rissardo JP, Leite MTB, Silveira JOF, Jauris PGM, Arend J, Kegler A, Royes LFF, Fighera MR. Course and prognosis of adult-onset epilepsy in Brazil: A cohort study. Epilepsy Behav. 2020 Apr;105:106969. doi: 10.1016/j.yebeh.2020.106969. Epub 2020 Feb 26. PMID: 32113113.

Provide a paragraph about the limitations of the study.

Review the references according to the instruction for authors. Also, review the misspellings in the reference list.

Round 2

Reviewer 2 Report

Comments and Suggestions for Authors

Satisfactory